

# The Improved Comparative Reactivity Method (ICRM): measurements of OH reactivity at high-NO$_X$ conditions in ambient air

**Wenjie Wang[1,3#], Jipeng Qi[1,2#], Jun Zhou[1,2], Bin Yuan[1,2*], Yuwen Peng[1,2], Sihang Wang[1,2], Jonathan Williams[4], Vinayak Sinha[5], Min Shao[1,2]**

[1]Institute for Environmental and Climate Research, Jinan University, Guangzhou 511443, China.

[2]Guangdong-Hongkong-Macau Joint Laboratory of Collaborative Innovation for Environmental Quality, Guangzhou 511443, China.

[3]Multiphase Chemistry Department, Max Planck Institute for Chemistry, Mainz 55128, Germany.

[4]Atmospheric Chemistry Department, Max Planck Institute for Chemistry, Mainz 55128, Germany.

[5]Department of Earth and Environmental Sciences, Indian Institute of Science Education and Research (IISER), Mohali 140306, India.

#W.J.W. and J.P.Q. contributed equally to this work.

*Correspondence to: byuan@jnu.edu.cn





**Abstract**

The comparative reactivity method (CRM) has been developed more than a decade to measure OH reactivity (i.e. OH loss frequency) in both laboratory and field studies. However, accurate OH reactivity quantification remains challenging under real ambient condition, especially for OH reactivity measurements in high-$NO_X$ (e.g. >10 ppbv) environments, as ambient NO enhance regeneration of OH radicals in the CRM reactor. To resolve this problem, we design a new improved CRM reactor (ICRM) and add NO into the system continuously, so that the $HO_2$ radical concentration is suppressed. We confirmed the appropriate level of NO by determining the maximum decrease in the pyrrole level caused by regenerated OH radicals from $NO + HO_2$. VOC-induced $RO_2$ radicals in the ICRM reactor were also found to react with NO, which lead to the re-generation of OH radicals thus the underestimation of OH reactivity. This effect was quantified by the calibration of representative VOC species at different NO levels, and the correction coefficients obtained were used to correct the measured OH reactivity. All these efforts resulted in reducing the uncertainty of the NO-artifact correction by at least an order of magnitude compared to the original CRM system. Additionally, these technological improvements also considerably reduced the systematic errors from pyrrole photolysis that exists in the original system. A new operation mode was proposed for ICRM, which is able to avoid the interference resulting from OH radicals produced by photolysis of residual humidity and save time for ambient measurement. The ICRM system was employed in a field campaign to measure OH reactivity and performed well with ambient NO levels ranged from 0 to 50 ppbv, which were typically observed in urban and suburban atmosphere.



## 1 Introduction


The hydroxyl radical (OH) is the most important oxidant in the daytime
troposphere. It initiates the chemically removal of primary gaseous pollutants and in
turn produces a host of secondary pollutants (A. Hofzumahaus, 1991; Atkinson, 2000;
Roger Atkinson, 2003). The OH reactivity is defined as the sum of all OH reactive trace
gas concentrations multiplied by their respective reaction rate coefficients with OH, as
shown in Eq. 1. The OH reactivity is a fundamental property of the atmosphere in that
it defines the overall loss frequency of OH radicals and hence the lifetime of OH. As
such it is a useful atmospheric parameter for evaluating the level of reactive pollutants
and it is closely related to atmospheric oxidation capacity and formation of secondary
pollutants including ozone and secondary aerosols (Sinha et al., 2012; Yang et al., 2016;
Pfannerstill et al., 2019).
$$R_{OH} = \sum_i^n k_{VOC_i}[VOC_i] + k_{CO}[CO] + k_{NO_2}[NO_2] + k_{SO_2}[SO_2] + k_{O_3}[O_3] \quad (1)$$

Equation 1 defines the OH reactivity where $R_{OH}$ is the total OH reactivity, $k_{CO}$,
$k_{NO}$, $k_{NO_2}$, $k_{SO_2}$, $k_{O_3}$, and $k_{VOC_i}$ represent the reaction rate coefficients between OH
radicals and CO, NO, $NO_2$, $SO_2$, $O_3$, and volatile organic compounds (VOCs) species
$i$, respectively. [CO], [NO], [$NO_2$], [$SO_2$], [$O_3$], and [$VOC_i$] are the concentrations of
CO, NO, $NO_2$, $SO_2$, $O_3$, and VOCs species $i$, respectively.
Currently, two general methods are used to measure OH reactivity: (1) direct
measurements of OH decay rates by laser-induced fluorescence (LIF) technique; (2)
measuring the relative change of a reference substance with and without ambient air
present by the comparative reactivity method (CRM). The LIF based technology has
been used to measure OH reactivity in a variety of different environments and has
provided many new insights into the budget of OH reactivity (Kovacs et al., 2001;
Kovacs et al., 2003; Sadanaga et al., 2004; Sadanaga et al., 2005; Ingham et al., 2009;
Lou et al., 2010). However, the cost, complexity, and large size of LIF systems are
deterrents to the widespread deployment for field measurements (Sinha et al., 2008).
Such detector systems need to be built and cannot be bought directly from a supplier.



The CRM method measures OH decay rate indirectly by using the relative reaction rate
of a reference substance (pyrrole) with self-generated OH radicals in the presence and
absence of ambient air. The reference substance can be measured by an online
instrument, such as proton transfer reaction mass spectrometry (PTR-MS) (Sinha et al.,
2008; Sinha et al., 2009; Kumar et al., 2014) or a gas chromatograph (Nolscher et al.,
2012a; Praplan et al., 2017a; Praplan et al., 2019b). The CRM technique has proven to
be a useful supplementary technique to measure the total OH reactivity in ambient air,
with a more economical and portable setup than the LIF-based systems. Based on inter-
comparison between various OH reactivity techniques in the SAPHIR chamber, the LIF
type-instruments are generally more sensitive and less noisy than CRM instruments
(Fuchs et al., 2017).
The CRM approach has been applied to numerous field campaigns in recent years
to measure OH reactivity (Dolgorouky et al., 2012; Nölscher et al., 2014; Michoud et
al., 2015; Kim et al., 2016; Zannoni et al., 2016; Praplan et al., 2017b; Yang et al., 2017a;
Zannoni et al., 2017; Kumar et al., 2018; Pfannerstill et al., 2018; Pfannerstill et al.,
2019; Praplan et al., 2019a). However, this method is not suitable for the environment
with high-level $NO_X$, due to the pen-ray mercury lamp used to generate OH radicals in
CRM system also generates approximately equivalent amounts of $HO_2$ radicals that
may react with sampled NO to produce additional OH radicals (Sinha et al., 2008; Yang
et al., 2017a), which cause an enhanced consumption of pyrrole in the CRM system and
result in an underestimation of OH reactivity in sampled ambient air when NO exceeds
certain levels (Sinha et al., 2008). This NO interference prevents the CRM method from
providing high-quality data in emission exhausts and urban areas with high NO levels.
As a result, applications of the CRM method have been generally restricted to high
reactivity/low $NO_X$ environments, including forests (Sinha et al., 2010; Kim et al., 2011;
Nolscher et al., 2012b; Praplan et al., 2019b; Pfannerstill et al., 2020), moderately
polluted cities (NO < 10 ppb) (Sinha et al., 2008; Praplan et al., 2017b), pristine marine
environments (Sinha et al., 2012; Zannoni et al., 2015), emission sources (e.g. gasoline
evaporation) (Wu et al., 2015), branch cuvette studies (Nölscher et al., 2013), and





chamber studies (Nölscher et al., 2014) with little or no NOx present. One solution to
this issue is to deliberately remove NO, before the sampled air is introduced into the
reactor. However, the present technology is not able to remove NO selectively without
affecting other reactive species (i.e., VOCs). The effect of NO on measured OH
reactivity can be quantified by NO-correction experiments and the resulting correction
curve applied to adjust ambient measurements according to simultaneously measured
NO levels (Hansen et al., 2015; Yang et al., 2017a). However, the uncertainty of
measured OH reactivity due to NO correction increases with NO concentration (Hansen
et al., 2015; Michoud et al., 2015). Hansen et al. (2015) reported that the total
uncertainty increases by up to a factor of 3 at NOx mixing ratios higher than 40 ppbv.
Therefore, it calls for an improvement of the traditional CRM reactor for accurately
quantifying OH reactivity at high NOx conditions. In addition to the NO effect,
photolysis of pyrrole and VOCs, and the humidity difference between zero air and
ambient air also influence measured OH reactivity (Sinha et al., 2008; Hansen et al.,
2015; Zannoni et al., 2015).
The main purpose of this study is to improve the original CRM system to make it
suitable for using in high-NOx conditions. We modified the structure of the original
CRM glass reactor and add a certain amount of NO into the system to remove the
generated but unwanted $HO_2$ radicals. We further characterized the improved CRM
(ICRM) system by quantitatively evaluating the effect of the reaction of sample VOC-
induced $RO_2$ with NO on measured OH reactivity. Additionally, the interference of
pyrrole photolysis was also systematically evaluated. Finally, the ICRM system was
deployed to measure OH reactivity under high-NOx conditions (0-50 ppbv) during a
field campaign in the Pearl River Delta region of China.

## 2 Experimental and Methodology

### 2.1 The original CRM reactor

The schematic of the original CRM reactor is shown in Fig. 1a. Gas-phase pyrrole
mixed with zero air or ambient air is introduced through arm C at a constant flow. Arm



A consists of a pen-ray spectral mercury lamp, over which nitrogen (humidified or dry)
is passed through arm B at a constant flow rate. When humidified nitrogen is flowing
and the mercury lamp is turned on, $H_2O$ is photolyzed into OH and H radicals by the
mercury lamp at 184.9 nm. The total air flow in the reactor exits through arm F and the
concentration of pyrrole is monitored with a PTR-MS. A detailed description of the
original CRM method has been reported by Sinha et al. (2008).
Figure 1(b) shows the four work modes of the original CRM method (Sinha et al.,
2009). In C0 mode, the mercury lamp is turned off and high-purity dry nitrogen is
introduced into the reactor through arm B. Pyrrole is introduced into the reactor with
dry zero air through arm C. In C1 mode, the mercury lamp is turned on while everything
else remains the same as C0. Pyrrole concentration decreases during C1 mode due to
its photolysis reaction. In C2 mode, nitrogen flow in C1 is changed to humidified
nitrogen to generate OH radicals, and the pyrrole concentration decreases to C2. In the
final step, ambient air is introduced to the reactor through arm C in C3 mode. Based on
pseudo-first-order assumptions (i.e., [pyrrole] ≥ [OH]), total OH reactivity ($R_{OH}$) is
calculated as Eq. 2:
$$R_{OH} = C1 \times k_{pyr+OH} \times \frac{C3-C2}{C1-C3} \tag{2}$$

Where $k_{pyr+OH}$ is the rate coefficient for the reaction of pyrrole with OH radicals
($1.28 \times 10^{-10}$ $cm^3 \cdot molecule^{-1} \cdot s^{-1}$ (Dillon et al., 2012)), and C1, C2, and C3 represent
pyrrole concentrations at the corresponding steps described above, respectively.
In C2 and C3 mode, OH radicals are produced by the photolysis of water vapor at
atmospheric pressure as shown in R1. The generated H radicals from R1 reacts with $O_2$
of zero air or ambient air to generate $HO_2$ radicals, as shown in R2. When NO is present
in the sampled ambient air, it can recycle OH by reacting with $HO_2$ (R3).
$$H_2O \xrightarrow{184.9nm} OH + H \tag{R1}$$

$$H + O_2 + M \rightarrow OH + H \tag{R2}$$

$$NO + HO_2 \rightarrow OH + NO_2 \tag{R3}$$

An underlying assumption of the CRM approach is that the influence of the species
in ambient air on OH radicals in the reactor is ignorable. However, the additional OH





radicals produced via R3 can react with pyrrole to cause an additional decrease in
pyrrole relative to C2 mode, thus lead to the underestimation of OH reactivity.

**2.2 The improved CRM reactor**

In order to remove the interference of the reaction of $HO_2$ radicals with NO as
discussed above, we modified the pipe structure of the original CRM reactor (Fig. 1c).
We kept the length and volume of the glass reactor of the ICRM system similar to the
original CRM system, but added a branch inlet G (1/4 inch OD glass; length 3 cm) in
arm A to introduce steam of NO standard (Air Liquide; stated uncertainty 3%; 10.8
ppmv) mixed with zero air at a constant flow of 120 ml·min$^{-1}$. The typical flow rate
inside the ICRM reactor is approximately 660 ml·min$^{-1}$. The flow rate of nitrogen (Air
Liquide; 99.9995% purity) through arm B is 250 ml·min$^{-1}$. The input pyrrole (Linde
Spectra Environment Gases; stated uncertainty 5%; 5.37 ppm) flow rate is 2.5 ml·min$^{-}$
$^1$. The total flow rate of pyrrole and zero air (Air Liquide; 99.9995% purity) through
arm C is 290 ml·min$^{-1}$. With this modified structure of arm A, the $HO_2$ radicals,
produced by the reaction of the generated H radicals near the mercury lamp and $O_2$ in
introduced zero air, were converted to OH radicals by reacting with NO in the
downstream of arm G. The interference induced by R3 can then be eliminated.
Arm A consists of one 1/2 inch OD (ID: 0.62 cm, length: 7 cm) glass tube and one
1/4 inch OD (ID: 0.32 cm, length: 5 cm) glass tube. The purpose of this structure is to
ensure that the UV light is mostly confined within a 1/2 inch OD glass tube of arm A,
as the diameter of arm A goes from wide to slender. The new structure of arm A leads
to lower OH concentrations (decreased by approximately 50%) passing into reactor due
to wall loss, OH radicals produced from the reaction of $HO_2$ radicals with NO can
partially compensate for this loss.

**2.3 The detection of pyrrole by PTR-MS**

The accuracy of pyrrole measurement is critical in determining OH reactivity for
CRM method. Here we used PTR-MS to detect pyrrole concentration. With a proton
affinity greater than water (Pyrrole: 209.2 kcal·mol$^{-1}$; Water: 165.2 kcal·mol$^{-1}$) (Sinha



et al., 2008), pyrrole is chemically ionized by proton transfer with $H_3O^+$ ions and the
product ions are detected using a quadrupole mass spectrometer. As highlighted by
Sinha et al. (2009), the sensitivity of PTR-MS instruments toward pyrrole is dependent
on humidity, and the pyrrole signal must be carefully calibrated for relative humidity
changes within the CRM reactor. The approach described by de Gouw and Warneke
(2007) was employed in this study to account for the effect of ion source and humidity
on the sensitivity of PTR-MS toward pyrrole (de Gouw et al., 2007). This approach
involves normalization of the pyrrole signal to a sum of reagent ion signals ($H_3O^+ + X_R$
$\times\ H_3O^+ \cdot H_2O$) that leads to a normalized signal for pyrrole that is independent of
humidity. $X_R$, a scaling factor for the $H_3O^+ \cdot H_2O$ signal, is determined experimentally
by measuring the pyrrole signal from a standard mixture under different humidity
conditions. In this study, a relatively higher electric field parameter of the drift tube (i.e.
E/N) value of 153 Td was used to measure pyrrole, which can minimize the humidity
effect from water clusters in the PTR-MS instrument. As shown in SI, the best estimate
for $X_R$ parameter was determined to be zero (Fig. S1), indicating negligible role for
$H_3O^+ \cdot H_2O$ in pyrrole detection by PTR-MS in this study.
**2.4 Other instruments of the ambient measurement campaign**

In order to test and validate the ICRM system for OH reactivity measurements

under high NO concentrations, we conducted field measurements of OH reactivity at a
receptor site in the Pearl River Delta (PRD) region of China (Yang et al., 2017b; Tan et
al., 2019). Meanwhile, non-methane hydrocarbons (NMHCs) and oxygenated volatile
organic compounds (OVOCs) were also measured by online gas chromatograph mass
spectrometer and flame ionization detector (GC-MS/FID) (Wang et al., 2014a) and
proton transfer reaction time-of-flight mass spectrometry (PTR-TOF-MS) (Yuan et al.,
2017), respectively (Table S1). Inorganic trace gases, including CO, $NO_2$, NO, $SO_2$ and
$O_3$, were measured by Thermofisher 48i CO analyzer, 2B Technologies Model 405nm
$NO_X$ analyzer, Thermofisher 42i $NO_X$ analyzer, Thermofisher 43i $SO_2$ analyzer, and
Thermofisher 49i $O_3$ analyzer, respectively. Detailed descriptions of these systems can
be found in previous studies (Wang et al., 2014b; Birks et al., 2018).


**2.5. Zero dimensional box model**


To test our understanding of the chemical processes occurring inside the ICRM
reactor, results from laboratory experiments were compared with simulation from zero-
dimensional (0-D) box model. The MCM v3.3.1 (Wolfe et al., 2016) was used as
chemical mechanism in the box model. The use of the detailed mechanism aims at better
representing the chemistry of peroxy radicals. In the box model, the initial
concentrations of OH, $HO_2$, pyrrole, VOCs, CO, and NO were supplied, and the time-
dependent variations of different compounds in the reactor are simulated. The initial
concentrations of OH (4 ppbv), $HO_2$ (4 ppbv), and pyrrole (21 ppbv) are determined
based on results from our experiments. The residence time in the reactor was about 11
s according to the volume of the reactor (120 $cm^3$) and the flow of introduced gases
(660 ml·$min^{-1}$). An MCM subset was extracted for inorganic reactions, and reactions
for propane, propene, and toluene. $C_3H_5O_2$ was used as a surrogate for the peroxy
radicals from pyrrole, as the degradation of pyrrole is not included in MCM.

**3 Results and discussion**


**3.1 Determination of the amount of NO addition**


In addition to $HO_2$ produced from the reaction of H radicals with $O_2$, $RO_2$ produced
from the reaction of pyrrole with OH also react with NO to recycle $HO_2$ and OH (R4-
R6), and consume pyrrole. In order to eliminate the effect of $HO_2$ and $RO_2$ radicals, NO
supply with an appropriate concentration through arm G is needed. We optimized NO
concentration by testing the dependence of the change of the pyrrole concentrations on
the concentration of NO introduced through arm G (as described below and Fig. 2).
During the experiment, the pyrrole concentration in the C1 mode (where $N_2$ and zero
air were humidified and mercury lamp is turned off) was 22 ppbv, which decreased to
18 ppbv when the mercury lamp was turned on at 0 ppbv NO, implying that the
generated OH radicals depleted ~ 4 ppbv pyrrole. We varied the NO concentrations
mixed with the zero air entering arm G which resulted in NO concentrations in the
reactor ranged from 0 to 150 ppbv, and found out the appropriate NO level to consume



all $HO_2$ and $RO_2$ produced in the glass reactor. NO was mixed with zero air rather than
nitrogen, as oxygen in zero air can transform H radicals in arm A to $HO_2$ radicals.
Pyrrole concentration decreases with the increase of NO concentrations, reaching a
minimum when NO concentration is circa $40 \sim 50$ ppbv, and increased again when NO
concentration exceeds 50 ppbv.

$$RH + OH + O_2 \rightarrow RO_2 + H_2O \qquad\qquad \text{R4}$$

$$RO_2 + NO + O_2 \rightarrow HO_2 + NO_2 + carbonyls \qquad \text{R5}$$

$$NO + HO_2 \rightarrow OH + NO_2 \qquad\qquad \text{R6}$$

$$RO_2 + HO_2 \rightarrow peroxide \qquad\qquad \text{R7}$$

where RH represents pyrrole in the reactor or introduced ambient VOCs into the reactor.

The NO addition experiments are simulated in the box model. The simulated

pyrrole concentrations as a function of NO concentration is consistent with laboratory
experiments: with pyrrole concentrations decreasing at first and then increasing (Fig.
S2). When NO is not present in the reactor, the self-reactions of peroxy radicals
($HO_2$+$HO_2$, $HO_2$+$RO_2$) dominate the sink of $HO_2$ and $RO_2$ (Fig. S3). As NO is
introduced into the reactor, the reaction of NO with $HO_2$ or $RO_2$ competes with the self-
reactions of peroxy radicals. With more NO introduced, the produced OH radicals from
the reaction of $HO_2$ with NO increase, leading to the decrease of pyrrole concentration
(Fig. S3). As the NO concentration exceeds 50 ppbv, pyrrole concentrations increase
again, due to the large excess NO competes with pyrrole for reaction with OH radicals.
The remaining NO concentration outflowing from the reactor increases with the
introduced NO concentrations (Fig. S2), indicating that excessive NO is needed to
compete with the self-reactions of peroxy radicals. Based on laboratory measurement,
the remaining NO concentration outflowing from the reactor is ~18 ppbv when the
introduced NO concentration at 50 ppbv. The laboratory measurements and simulated
results both suggest that $40 \sim 50$ ppbv is the lowest NO concentration needed to
transform $HO_2$ and $RO_2$ to OH to the largest extent. The higher introduced NO
concentration had a negligible effect on the increase in OH production from $HO_2$ and
$RO_2$. Thus, we introduce 50 ppbv NO concentration into the ICRM reactor in the



experiments in this study. Under this optimized condition, the pyrrole concentration
decreased to 12.3 ppbv. The concentration of pyrrole in this scenario is regarded as the
C2 mode for ICRM system. It worth noting that the determined NO concentration can
vary slightly as OH generation performance changes (e.g. humidity change in the region
of the pen-ray mercury lamp).
Under the determined optimal NO level through arm G, it is necessary to ensure
that the OH production from $HO_2$ and pyrrole-induced $RO_2$ will not manifest itself
when ambient NO is introduced through arm C. For this purpose, we compared
measured and true OH reactivity of NO by passing a series of NO concentrations (0 ~
160 ppbv) mixed with zero air through arm C into the reactor (Figure 3). In this test, no
other reactive gases were introduced into the system except NO. Measured OH
reactivity of NO agreed well with the corresponding calculated values, indicating that
$HO_2$ radicals have been fully consumed, and pyrrole peroxy radicals were effectively
converted to carbonyls and nitrates by NO introduced through arm G.
**3.2 Calibration for OH reactivity of VOCs**
Several reactive VOC species were used to validate and calibrate the ICRM system,
including propane, propene, toluene, and mixed gases including 16 VOC species
(acetaldehyde, methanol, ethanol, isoprene, acetone, acetonitrile, methyl vinyl ketone,
methyl ethyl ketone, benzene, toluene, o-xylene, α-pinene, 1,2,4-trimethylbenzene,
phenol, m-cresol, naphthalene). These VOC species were introduced into the system
through arm C at various reactivity (0 ~ 80 $s^{-1}$). Figure 4 (a) presents the plots of the
measured ($R_{meas}$, $s^{-1}$) versus true OH reactivity ($R_{true}$, $s^{-1}$) of these species. $R_{meas}$ is lower
than $R_{true}$ for almost all species with the slopes of linear fittings ranging from 0.70 to
0.74. The slopes of propane, propene, toluene, and mixed gases are 0.70, 0.74, 0.72,
and 0.72, respectively. The OH reactivity calibration of $SO_2$ and CO indicates that the
linear fitting slope of $R_{meas}$ versus $R_{true}$ is 0.33 and 0.41, respectively (Fig. 4 b), which
is lower than that of VOCs.
Equation 2 is valid only under near pseudo-first-order conditions (i.e. when
[pyrrole] >> [OH]). In this study, the [pyrrole] to [OH] ratio is set at 2.5, which will


cause significant systematic errors. We plot the calculated reactivity, obtained by
applying Eq. 2 to the numerical simulations of the pyrrole concentration (C2 and C3)
at [Pyrrole]/[OH] = 2.5 after OH had reacted to zero, versus the true reactivity. The
correction curve indicates that the calculated reactivity underestimates the true
reactivity by about 5%. After considering this interference, the slope of calibration
shown in Fig. 4 (a) and 4(b) decreased to 0.66 ~ 0.70 for VOCs, 0.31 for $SO_2$ and 0.39
for CO, respectively. Therefore, the deviation of pseudo-first-order conditions cannot
explain the calibrated slopes for VOCs, CO and $SO_2$ being lower than one.
The lower calibrated slopes for VOCs than one can be related to secondary
chemistry of VOC-generated $RO_2$ radicals with NO. When more VOCs are introduced
into the reactor, additional $RO_2$ radicals produced from the reaction of VOCs with OH
will react with excessive NO in the reactor thus increase the recycled OH (R4-R6). The
recycled OH from $RO_2$ will deplete pyrrole thus leading to a $R_{meas}$ lower than the $R_{true}$.
We deduce that this is the reason for the linear fitting slopes in Fig. 4 lower than one.
For specific VOC species, the decrease in pyrrole concentration due to recycling OH
depends on the true OH reactivity of VOCs, NO concentrations and the efficiency of
organic nitrate production ($RO_2$ + NO → $RONO_2$) in this system. The consistency in
the linear fitting slopes of different VOC species indicate that the $RO_2$ + NO reactions
for the investigated VOCs are similar. This is in agreement with the simulated results
(Fig. S7). The lower fitting slope of $SO_2$ and CO than that of VOCs is because $SO_2$ and
CO react with OH to produce $HO_2$, which has higher efficiency to produce OH by
reacting with NO than $RO_2$ that goes through two steps ($RO_2$→$HO_2$, and $HO_2$→OH).
Here, we define the linear fitting slopes in Fig. 4 as correction coefficients with regard
to the calibration for OH reactivity of VOC, CO and $SO_2$ (characterized by $\alpha_{VOC}$, $\alpha_{CO}$
and $\alpha_{SO2}$) at ambient NO = 0 ppbv.
To further evaluate the performance of the ICRM system with elevated $NO_X$
concentrations in ambient air, a series of NO concentrations were introduced into the
reactor through arm C both with constant reactivity from different VOC species and
with different reactivities provided by the same species. Of all experimental conditions,



the $R'_{meas}$ (The $R'_{meas}$ is defined as the corrected $R_{meas}$ by correction coefficient $\alpha_{VOC}$)
was observed to decrease with increased NO concentration (Fig. S5), and thus the
difference between $R_{true}$ and $R'_{meas}$ ($R_{true} - R'_{meas}$) increases with increased NO
concentrations for the four VOC standard gases (Fig. 5a and b). This is because the
reaction rate of $RO_2$ with NO increases with NO concentrations leading to enhancement
of the recycled OH. The linear fitting slopes of ($R_{true} - R'_{meas}$) versus NO concentrations
for different VOC species and different reactivity levels range from 0.10 to 0.22 $s^{-1}$
$ppbv^{-1}$. Similar to VOCs, the difference between $R_{true}$ and $R'_{meas}$ ($R_{true} - R'_{meas}$) also
increases with NO concentrations for CO and $SO_2$ with slope of 0.11 and 0.10
respectively (The $R'_{meas}$ is defined as the corrected $R_{meas}$ by correction coefficient $\alpha_{CO}$
and $\alpha_{SO_2}$) (Fig. 5 c). However, the difference of NO effects between VOCs and CO (and
$SO_2$) as shown in Fig. 4 and Fig. 5 has not been reported in previous studies about CRM
system.

Previous studies have reported that NO had a large effect on the difference between

$R_{true}$ and $R_{meas}$ in the CRM systems (Note that the $R_{meas}$ is not corrected in previous
studies) (Hansen et al., 2015; Michoud et al., 2015; Yang et al., 2017b). This NO-effect
is not only due to the reaction of $HO_2$ with NO, but also due to the reaction of pyrrole-
produced and VOC-produced $RO_2$ with NO. Figure S6 compares the effect of NO on
($R_{true} - R_{meas}$) in the original CRM system (reported by previous studies) with that in the
ICRM system (this study). Far larger NO effects were reported in the original CRM
system than in the ICRM system. For example, the presence of ambient NO at 50 ppbv
leads to $R_{meas}$ lower than the $R_{true}$ by $70 \sim 240$ $s^{-1}$, at least an order of magnitude higher
than the NO artifact in the ICRM system, which leads to $R'_{meas}$ lower than the $R_{true}$ by
$5 \sim 13$ $s^{-1}$. This is because both $HO_2$ and pyrrole-induced $RO_2$ are fully removed by the
introduced NO in advance in the ICRM system, thus the remaining influencing factor
is the reaction of ambient VOCs-induced $RO_2$ with NO. The uncertainty due to the NO-
artifact correction in the ICRM system was predicted to be far lower than that of the
original CRM system, as the absolute change of OH reactivity caused by NO is reduced
by removing $HO_2$ and pyrrole-induced $RO_2$. Despite the ICRM system not being able



to remove the NO effect entirely, it does lead to a significant decrease in the uncertainty
of the NO-artifact correction.

Due to the different behaviors of VOCs, $SO_2$ and CO at high NO conditions, in

order to get accurate OH reactivity, it is necessary to conduct NO-correction for VOCs,
$SO_2$, and CO individually. Note that this issue may also present in the original CRM
system, but it was ignored in previous studies. For the ICRM system, we use the
following formula to determine the true OH reactivity of VOCs:
$R_{meas} = R_{true\ NO+NO_2} + R_{true\ O_3} + \alpha_{CO}(R_{true\ co} - \beta_{CO}[NO]) + \alpha_{SO_2}(R_{true\ SO_2} -$

$\beta_{SO_2}[NO]) + \alpha_{VOC}(R_{true\ voc} - \beta_{VOC}[NO])$ (3)

$R_{true\ VOC} = \frac{1}{\alpha_{VOC}}(R_{meas} - R_{true\ NO+NO_2} - R_{true\ O_3} + \alpha_{co}\beta_{co}[NO] +$

$\alpha_{SO_2}\beta_{SO_2}[NO] + \alpha_{voc}\beta_{VOC}[NO] - \alpha_{CO}R_{true\ co} - \alpha_{SO_2}R_{true\ SO_2})$    (4)

where $R_{meas}$ is the measured OH reactivity by the ICRM system as defined above.

The $R_{true\ VOC}$ is the true OH reactivity of VOCs. $R_{i\ true}$ was calculated from measured
concentrations of species $i$ ($i$=NO, $NO_2$, $O_3$, $SO_2$ and CO) multiplied by the rate
coefficient of the reaction of species $i$ with OH. The $R_{meas}$ and $R_{true}$ of $NO_X$ (=NO+$NO_2$)
was close to 1:1 as shown in Fig. 3 and S4. $\alpha_{CO}$, $\alpha_{SO_2}$, and $\alpha_{VOC}$ are the correction
coefficients with regard to the calibration for OH reactivity of CO, $SO_2$ and VOC at
ambient NO = 0 ppbv, respectively. Note that the $\alpha_{VOC}$ is mean slope of Fig. 4 a $\beta_{CO}$,
$\beta_{SO_2}$, and $\beta_{VOC}$ are the correction coefficients which regard to the effect of ambient
NO on ($R_{true} - R'_{meas}$). Note that the $\beta_{VOC}$ is mean slope of Fig. 5 a and b. After getting
$R_{true\ VOC}$, the total OH reactivity ($R_{tot}$) was then calculated as the summation of
$R_{true\ VOC}$, $R_{true\ NO+NO_2}$, $R_{true\ O_3}$, $R_{true\ SO_2}$, and $R_{true\ co}$:

$R_{tot} = R_{true\ VOC} + R_{true\ NO} + R_{O_3} + R_{true\ NO2} + R_{true\ CO} + R_{true\ SO_2}$        (5)

**3.3 Additional potential interference related to NO addition**

In order to assess the extent of any additional interferences due to NO addition,

we further consider the following effects.

In arm A, the photolysis of $O_2$ introduced through arm G by the mercury lamp

produces $O_3$. Besides, the NO introduced through arm G reacts with $HO_2$ to generate





$NO_2$, which can also photolysis to generate NO and oxygen atoms, and subsequently
$O_3$. We monitored $O_3$ concentration through the arm F using an $O_3$ monitor. $O_3$
concentration flowing out of arm F was less than 5 ppbv, which has a negligible
influence on the pyrrole concentrations and the $R_{meas}$, considering the pyrrole+$O_3$
reaction rate constant $k_{O3+pyrrole}$ =$1.57\times10^{-17}$ cm$^3$ molecule$^{-1}$ s$^{-1}$ (Atkinson et al., 1984)
is several orders of magnitude slower than the pyrrole + OH reaction rate constant
($k_{pyrrole+OH}$ =$1.28\times10^{-10}$ cm$^3$ molecule$^{-1}$ s$^{-1}$). The ozone concentration was low, as excess
NO was introduced to the reactor and the remaining NO titrates $O_3$ back to $NO_2$.

In the C3 mode of the ICRM, sample ambient $O_3$ can react with the high-levels

NO in the reactor, which might interfere with $R_{meas}$. We characterize this interference
by introducing a series of $O_3$ concentrations into the reactor through arm C. As $O_3$
concentrations lower than 40 ppbv, $O_3$ has a negligible effect on OH reactivity (Fig. 6).
Interestingly, $R_{meas}$ first increases and then decrease with increasing $O_3$ concentrations.
The reaction rate coefficient of OH with $NO_2$ is slightly higher than with NO, which
are $1.2\times10^{-11}$ cm$^3$ molecule$^{-1}$ s$^{-1}$ and $1.0\times10^{-11}$ cm$^3$ molecule$^{-1}$ s$^{-1}$ at 298K, respectively
(Atkinson et al., 2004). With the increase of introduced $O_3$ concentration, higher $NO_2$
is produced, which causes an increase in $R_{meas}$. As NO is consumed completely by $O_3$,
excessive $O_3$ can further react with $NO_2$ to produce $NO_3$ radicals, which can deplete
pyrrole ($k$=$1.80 \times 10^{-10}$ cm$^3$ molecule$^{-1}$ s$^{-1}$) (Cabanas et al., 2004) and lead to the
decrease in $R_{meas}$. Overall, OH reactivity exhibited little change (< 2 s$^{-1}$) with the
increase of $O_3$ concentrations (0 ~ 160 ppbv), indicating that the introduced $O_3$ plays a
negligible role in $R_{meas}$. This is another advantage of ICRM compared with the original
CRM, which needs an ozone correction as the reaction of $O_3$ with $HO_2$ gives OH back
(Fuchs et al., 2017).

According to model simulation, the produced $NO_2$ from the reaction of NO with

$HO_2$ increases with introduced NO concentrations (Fig. S2). The produced $NO_2$ can
deplete OH (OH + $NO_2$ → $HNO_3$) and thereby lead to an increase in the pyrrole
concentration. When introduced NO with a concentration of 50 ppbv, the produced $NO_2$
was 25 ppbv, corresponding to 6.2 s$^{-1}$ OH reactivity (Fig. S2). However, this process





doesn't interfere with the $R_{meas}$ as the produced $NO_2$ is the same in both C2 and C3
modes leading to this effect canceling out in the two modes.
**3.4 Photolysis of pyrrole**
Photolysis of pyrrole in the CRM method introduces additional uncertainties and
complexity in the determination of OH reactivity (Sinha et al., 2008; Hansen et al.,
2015; Michoud et al., 2015; Zannoni et al., 2015). To investigate the effect of the ICRM
system on the interference from photolysis, we turn the mercury lamp off and on to test
the variation in pyrrole concentrations under dry conditions (no humidification).
Compared with the condition where the mercury lamp is turned off, pyrrole
concentrations decreased by < 3% after the mercury lamp was turned on (Fig. S8),
which caused $R_{meas}$ increase of 0.55 s$^{-1}$ when the $R_{true}$ was 20 s$^{-1}$. This result indicates
that the photolysis of pyrrole is weak enough to be negligible in the ICRM system. This
smaller photolysis of pyrrole closely relates to the improved design of reactor structure.
Arm A consists of two section of glass tube with 1/2 inch OD and 1/4 inch OD,
respectively (Fig. 1). UV light is mostly confined in 1/2 inch OD glass tube of arm A,
as the glass tube is constructed with decreasing diameter following the direction of gas
flow. This reduces the amount of UV light getting into the main reaction part of the
reactor. The improved structure of arm A leads to lower OH concentrations (decreased
by approximately 50%) passing into reactor due to wall loss, but the OH radicals
produced from the reaction of $HO_2$ radicals with NO can partially compensate for this
loss. In comparison, the pen-ray mercury lamp was very close to the main body of the
reactor in the original CRM reactor, to maximize the OH entering the reactor by
minimizing wall loss. However, this will lead to the photolysis of pyrrole, as high as
25% (Sinha et al., 2008; Hansen et al., 2015). The change of the structure of arm A also
ensures that the photolysis of $H_2O$, HONO, $NO_2$, and VOCs inside the ICRM reactor is
weaker than that in the original CRM system. In addition to our design change, previous
studies have reported that the photolysis of pyrrole can be also lowered to below 5% by
changing the UV mercury lamp position in the setup (Michoud et al., 2015; Zannoni et
al., 2015).


In the original CRM system, C1 instead of C0 is used as the initial amount of
pyrrole in order to avoid the interference of pyrrole photolysis. The C1 mode, where
dry $N_2$ and zero air are used meanwhile the mercury lamp is turned on, was measured
every 12 h for a duration of 2 h (Sinha et al., 2008; Hansen et al., 2015). The length of
the duration is necessary to reach the driest conditions possible to minimize residual
OH in the reactor. It should be noted that this procedure can result in an underestimation
of C1, as it is difficult to remove all trace amounts of water molecules from surfaces
and in nitrogen and zero air flowing through the reactor, which is able to produce extra
OH by photolysis (Hansen et al., 2015; Michoud et al., 2015; Zannoni et al., 2015). The
underestimation of C1 will result in an overestimation of OH reactivity. The
significantly smaller photolysis of pyrrole for the ICRM system allows us to measure
the C1 mode differently. Here, the condition that $N_2$ and zero air are humidified while
the mercury lamp is turned off is regarded as C1. The new C1 mode is able to avoid the
interference resulting from OH radicals produced by photolysis of residual humidity
since the mercury lamp is turned off and OH will not be produced. Besides, the C1
mode in ICRM is measured every 12 h for a duration of 15 min, which also saves time
compared with C1 mode in original CRM.
**3.5 Humidity difference between zero air and ambient air**
The variation of humidity between the C2 (wet zero air) and C3 (ambient air)
measurements could result in a change in OH production rate in the CRM reactor, which
in turn could lead to a C2 measurement not representative of the OH production rate
observed during the C3 measurement (Sinha et al., 2008; Dolgorouky et al., 2012).
Although the use of a catalytic converter or dynamic humidification of zero air can help
to reduce differences in humidity between C2 and C3 modes, small differences still
exist (Michoud et al., 2015). Besides, while catalytic converters can be used to generate
zero air with the same humidity as ambient air, these converters cannot remove NOx
species and thus are not suitable for OH reactivity measurements in urban and suburban
areas with high NOx conditions (Hansen et al., 2015).
To investigate the influence of humidity differences between C2 and C3 on the





$R_{meas}$ in the ICRM system, we test the response of pyrrole concentration to humidity by
introducing zero air with different humidities through arm C at mode C2. The ratio of
$H_3O^+(H_2O)$ to $H_3O^+$ (m37/m19) is selected to represent the level of different humidity.
Figure 7 (a) presents the dependence of pyrrole concentrations on m37/m19 at mode
C2. Pyrrole concentrations slightly decrease with the increase in m37/m19. It must be
noted that this dependence is not due to the humidity dependence of the PTR-MS
sensitivity toward pyrrole, but the change in OH production in the reactor, as the
normalization procedure of pyrrole signal described in Sect. 2.3 was applied to all
pyrrole measurements. The maximum difference of m37/m19 between C2 mode and
C3 mode is about 0.01 corresponding to RH changing by ~ 30% (Fig. 7 b), which lead
to pyrrole changed by ~ 0.26 ppbv and thus the $R_{meas}$ changed by ~1.9 s$^{-1}$ when the $R_{true}$
is 20 s$^{-1}$. This result indicates that the influence of humidity change on OH
concentrations and subsequently $R_{meas}$ cannot be ignored even though the structure of
arm A was improved to decrease the numbers of photons entering the main body of the
reactor. Therefore, humidity correction is needed to accurately $R_{meas}$. The humidity
difference between C2 and C3 mode can be corrected by the function derived from the
relationship between OH reactivity and m37/m19 (as shown in Fig. 7 a).
**3.6 Instrument performance in ambient measurements**
During the measurement, daily calibration was conducted by introducing a
constant concentration of various VOCs standards (propane, propene or toluene)
through arm C at C2 mode and determining the ratio of $R_{meas}$ to $R_{true}$ (i.e. $\alpha_{VOC}$). As shown
in Fig. S9, $R_{meas}$ / $R_{true}$ is relatively stable during the measurement, ranging from 0.60 to
0.70, implying this method has high stability, despite the structural differences of the
VOCs species introduced.
Figure 8 (a) presents a time series of $R_{tot}$, calculated OH reactivity ($R_{cal}$), and
ambient NO. $R_{tot}$ was acquired based on Eq. 5, and $R_{cal}$ is calculated by the sum of all
measured reactive trace gas concentrations multiplied by their respective reaction rate
coefficients with OH. The new system worked well even at high NO concentrations (>
20 ppbv). The average $R_{tot}$ for the entire campaign was 27 s$^{-1}$. The $R_{tot}$ is higher than the





$R_{cal}$ by 34% during the campaign, with larger differences observed in the morning and
at night than in the afternoon. As shown in Fig. 8(b), the $R_{tot}$ has an obvious diurnal
variation with higher levels at night and morning than that in the afternoon. This is
because air pollutants from anthropogenic emissions were accumulated at night and
morning due to lack of oxidative consumption, whereas were depleted rapidly during
the afternoon due to rising levels of oxidant, i.e. OH radicals. This diurnal pattern of
$R_{tot}$ is similar to that of the previous measurement results in the Pearl River Delta (Lou
et al., 2010; Yang et al., 2017a) and Beijing (Williams et al., 2016). Overall, the diurnal
variation of the $R_{true\ VOC}$ (calculated by Eq.4) is similar to that of the calculated OH
reactivity of inorganic gas (Fig. 8 b) and the concentration of $NO_X$ (Fig. 8 c). A
comparison between the $R_{tot}$ determined by the ICRM method and the laser-induced
fluorescence method will be of interest in future studies, particularly because LIF type
systems can also experience difficulties at high NO when OH decay rates are rapid.
Further discussions on the OH reactivity results of this campaign will be given in
another publication.

## 4 Conclusion

In this study, we presented an improved comparative reactivity method (ICRM)
which is suitable for measuring OH reactivity under high-NO conditions. The major
improvements of ICRM compared to the original CRM system are as follows:
(1)  The $HO_2$ and $RO_2$ radicals produced from H radicals reacting with $O_2$ and OH-
oxidation of pyrrole, respectively, were removed continuously to the largest extent. In
this study, 50 ppbv NO was inject into the ICRM reactor through an additional arm G
between arm A and the reactor. Under this NO level, the interference due to the reaction
of $HO_2$ and $RO_2$ from pyrrole with NO was minimized.
(2)  The OH recycling always happens to some extent when sampled VOCs are
introduced into the reactor in the presence of NO, causing the measured OH reactivity
($R_{meas}$) deviate from the true OH reactivity ($R_{true}$). We quantified this effect by
calibrating several representative VOC species, CO and $SO_2$ to obtain the slope of $R_{meas}$
versus $R_{true}$. Different VOC species produce similar slopes, which are significantly



higher than the slopes of CO and SO$_2$. Using the average value of the derived slopes of
the different species as a correction factor, we obtained the more accurate $R_{meas}$.
Additionally, the effect of ambient NO on the difference between $R_{true}$ and $R'_{meas}$ was
quantified.
(3) Transforming the structure of the glass reactor to reduce the amount of ultraviolet
light generated by the mercury lamp reaching the main body of the glass reactor. This
effort resulted in eliminating the interference of pyrrole photolysis existed in the
original system. Under this condition, the new C1 mode used was able to avoid the
interference resulting from OH radicals produced by photolysis of residual humidity
and save lots of time compared with the original C1 mode. The ICRM system was
employed in a field campaign to measure OH reactivity and performed well even if
ambient NO concentrations are high.

**Data availability**
The more detailed data can be provided by contacting the corresponding author.

**Author contributions**
WJW and BY came up the idea for the improved CRM. JPQ built the ICRM system
and performed data analysis. WJW, JPQ and BY wrote the manuscript, with
contributions from all other authors. YWP and SHW provided the PTR-TOF-MS and
PAMS data.

**Competing interests**
The author declares that there is no conflict of interest.

**Acknowledgment**
This work was supported by Key-Area Research and Development Program of
Guangdong Province (grant No. 2019B110206001), the National Natural Science
Foundation of China (grant No. 41877302), Guangdong Natural Science Funds for



Distinguished Young Scholar (grant No. 2018B030306037), the National Key R&D
Plan of China (grant No. 2019YFE0106300, 2018YFC0213904, 2016YFC0202206),
Guangdong Soft Science Research Program (2019B101001005), and Guangdong
Innovative and Entrepreneurial Research Team Program (grant No. 2016ZT06N263).
This work was also supported by Special Fund Project for Science and Technology
Innovation Strategy of Guangdong Province (Grant No.2019B121205004).



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



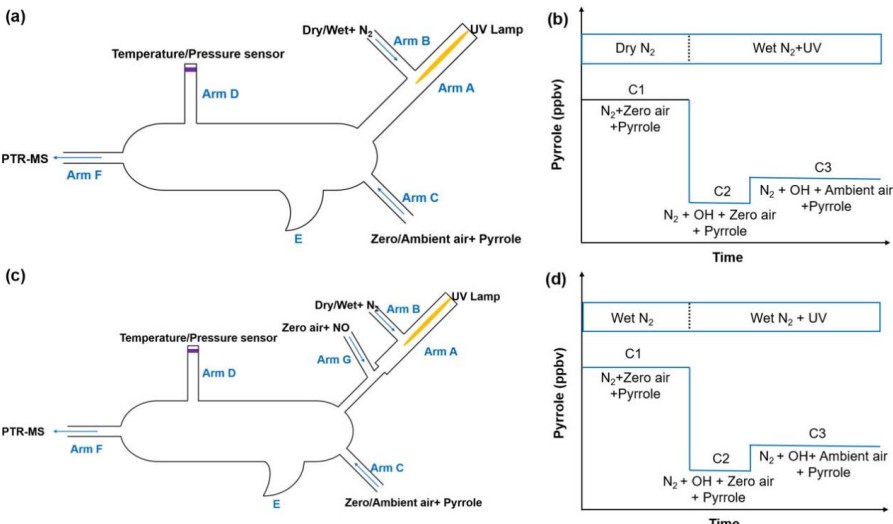


**Figure 1**. Schematic and work mode of the original CRM method (Sinha et al., 2008,

a and b) and the ICRM method (this study, c and d).



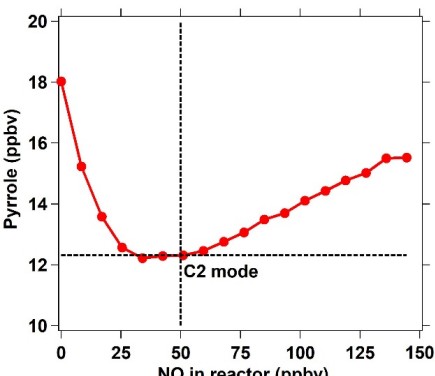


**Figure 2**. The response of pyrrole concentration to different NO concentrations introduced through arm G into the reactor. For the ICRM system, the C2 mode is corresponding to the pyrrole concentration = 12.31 ppbv at NO = 50 ppbv where the $HO_2$ radicals were removed constantly.





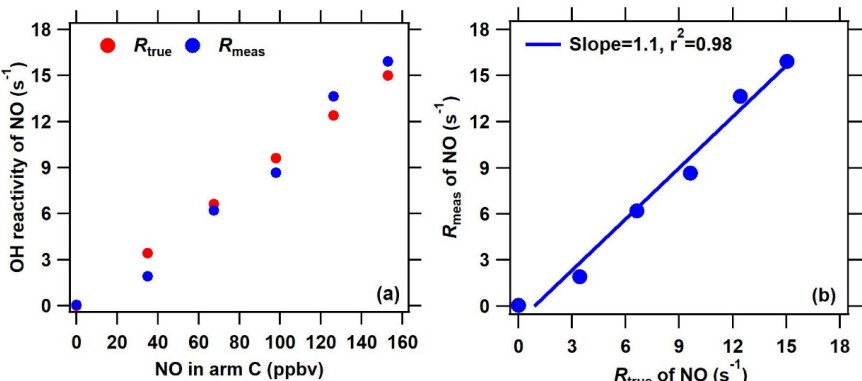

**Figure 3.** Comparison of measured and true OH reactivity of NO at different NO concentrations introduced through arm C. The measured OH reactivity of NO was calculated based on the new C2 mode shown in Fig. 2 in the ICRM system.



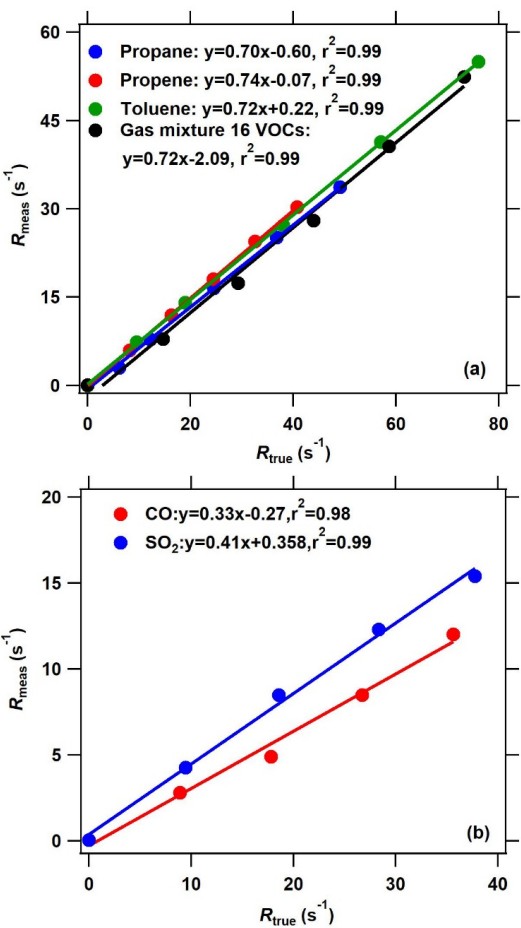

770

**Figure 4**. The OH reactivity calibration of the improved CRM system using different standard gases. **(a)** The calibrating results of organic species including propane, propene toluene, and mixture gases of 16 VOC species through arm C. **(b)** The calibrating results of inorganic species including CO and $SO_2$. The measured OH reactivity was calculated based on the C2 mode shown in Fig. 2 in the ICRM system.

776



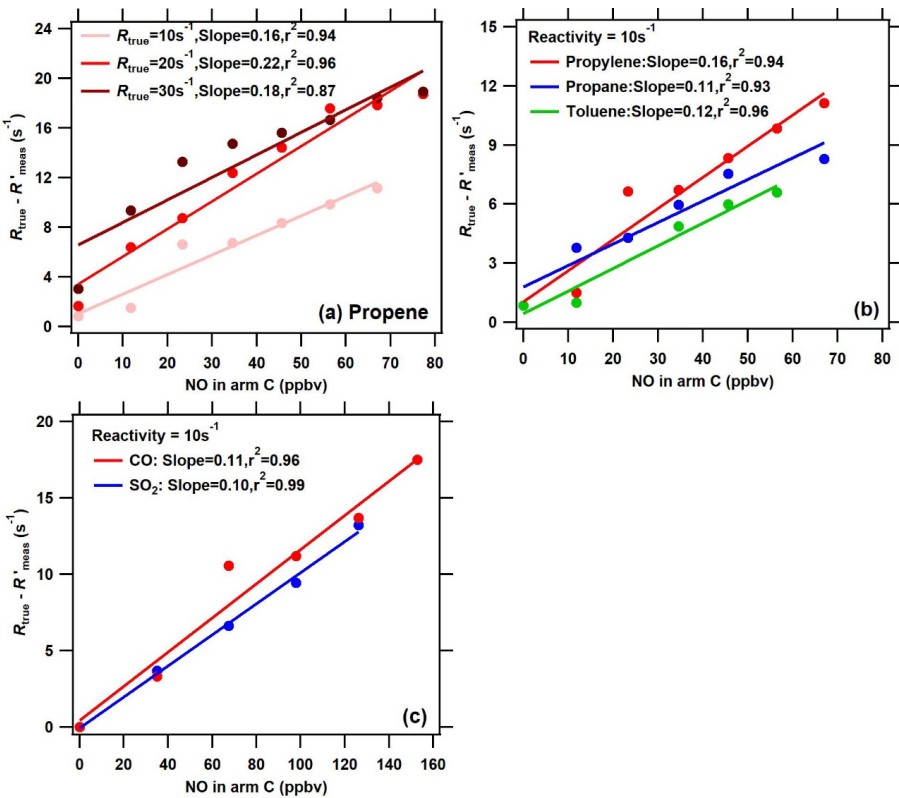

**Figure 5**. The difference between true OH reactivity ($R_{true}$) and the corrected measured

OH reactivity ($R'_{meas}$) using the calibration factor $\alpha_1$ ($R'_{meas} = (\frac{1}{\alpha_{VOC}} * R_{meas})$) as a

function of NO concentrations in arm C in the conditions of **(a)** different levels of VOCs

reactivity for the same species (propylene), **(b)** different VOCs species for the same

OH reactivity level (10 s$^{-1}$), and **(c)** different inorganic species (Red: CO; Blue: SO$_2$)

for the same OH reactivity level (10 s$^{-1}$). Note that NO, CO, SO$_2$, and VOCs were

introduced into the reactor through arm C.





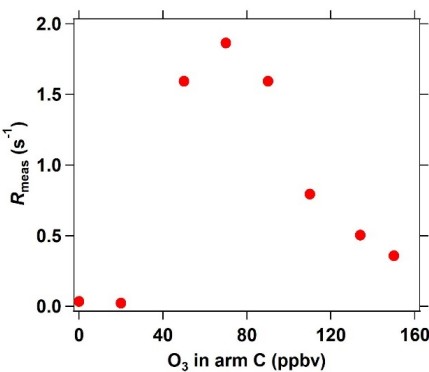

786

**Figure 6**. Interference of different $O_3$ concentration (introduced into the reactor through

arm C) on measured OH reactivity in the ICRM system.

789

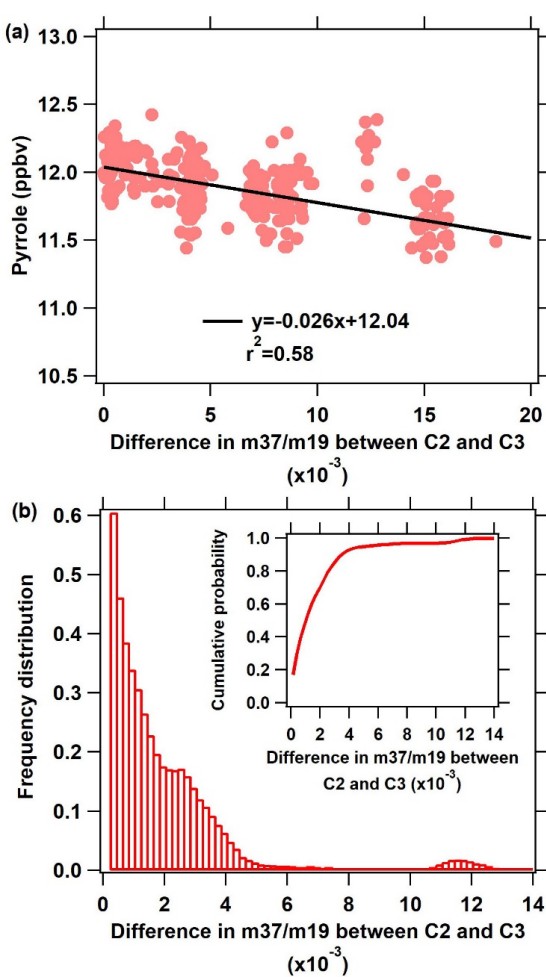

790

**Figure 7**. **(a)** Pyrrole concentration during zero air measurements (C2) as a function of

the difference in m37/m19 between C2 and C3 humidity indicator (m37/m19). **(b)**

Frequency distribution of the difference in m37/m19 between C2 and C3 during the

measurement.

795

796



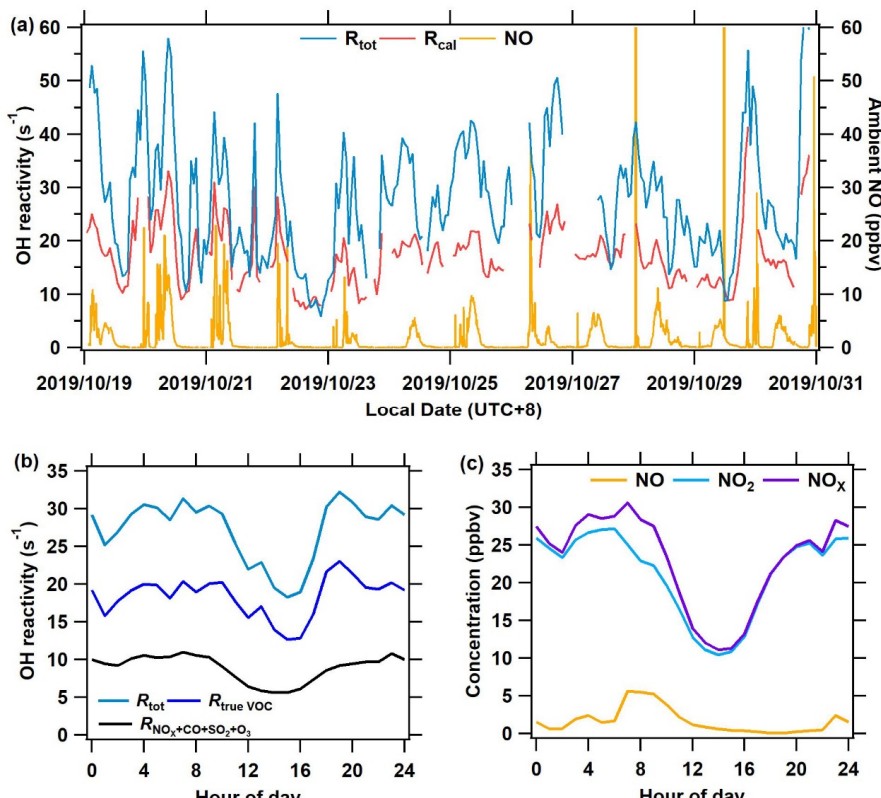

**Figure 8**. The measurement results of OH reactivity and ambient NO at the Heshan site

from October 19 to October 31, 2019. **(a)** The time series of total OH reactivity ($R_{tot}$),

calculated reactivity ($R_{cal}$), and ambient NO concentration; **(b)** Mean diurnal profile of

OH reactivity of $R_{true\ VOC}$, $R_{NO_X+CO+SO_2+O_3}$, and $R_{tot}$; **(c)** Mean diurnal profiles of

measured NO, $NO_2$, and $NO_X$.