# Peer review of "The Improved Comparative Reactivity Method (ICRM)"

_Atmospheric Measurement Techniques, 2020_

## Referee Comment (RC1) · Anonymous Referee #1 · 30 Dec 2020

The authors are presenting an innovative approach to address two well-known analytical artifacts in the CRM OH reactivity - the pyrrole photolysis and the OH recycling issues. This starts from improving the reactor, particularly, to prevent UV photon leaks and introduce excess NO.

The manuscript is easy to follow and has merits to the atmospheric chemistry community particularly who intends applying the CRM method for their field observations or using the dataset for the research. I would recommend the publication of this manuscript after addressing following points.

1) The reaction of CO with OH will produce HO2, which ends up recycling OH in the

excess NO environment. Please include the discussion in the manuscript.

2) It is certainly a good start to test the ICRM reactor with three VOCs but likely VOCs with various function groups required to evaluate. If the authors could suggest the list of compounds to be tested and provide justification, that would trigger follow up studies from other research groups.

—————————————————————

---

## Referee Comment (RC2) · Anonymous Referee #2 · 30 Dec 2020

The manuscript reported an improved CRM method to measure OH reactivity. The reactor of traditional CRM method was modified to suppress HO2 formation. The lab experiment results were promising that the measured and calculated kOH by trace gases have good linearity. The ambient measurements showed that the new ICRM method could be used under urban ambient air. The manuscript is well within the scope of AMT. I recommend publication after attention to the following comments.

General comments:

1. The reaction time of NO+HO2(or OH) in arm A should be clearly specified. HONO will be formed with no doubt and will it cause any interference with CRM?

[Figure]

2. In section 3.1, the NO addition measurement showed that Pyrrole concentration decreased to the minimum with NO around 40~50 ppbv (Fig.2). I think the increase of Pyrrole with NO is due to NO+OH reaction which lower the produced OH concentration.

3. In section 3.2 and Fig.4, the authors gave very promising dataset of the measured and calculated kOH. The linearity were all very good but the slope were not close to 1. Since the ICRM method introduced extra NO in the reactor, the cycling of OH-HO2-OH can not be avoid due to the reaction time in the reactor. The influence of initial HO2 was suppressed and at mean time the HOx cycling was enhanced. I would guess the slopes of CO and VOCs in Fig. 4 are related to this issue. Probably more VOCs should be tested before application in ambient air.

Specific comments:

Line 58: Better to include NO in the equation.

Line 131: Please specify the brand and type of the lamp, as well as its emission line.

Line 158: "An underlying assumption of the CRM approach is that the influence of the species in ambient air on OH radicals in the reactor is ignorable." The sentence is ambiguous. It is also useful to give the theoretical OH mixing ratio in the reactor here.

Line 181: Did the author try different structure (length, ID .etc) of arm A to get an optimal setup?

Line 250: It is better to include OH+NO reaction here. Is this reaction also include in the box model?

Line 283: the rate constant of OH+NO should be given here or in Fig.3, when calculated R-true of NO.

Line 404: The rate constant were quoted from Atkinson 2004, which is a well-known reference. I would suggest the authors also check the new evaluations or recommendations on JPL-2015 or IPUAC sources.

---

## Author Comment (AC1) · 23 Jan 2021

**Responses to Reviewers**

**Reviewer 1**

The authors are presenting an innovative approach to address two well-known recycling issues. This starts from improving the reactor, particularly, to prevent UV photon leaks and introduce excess NO.

The manuscript is easy to follow and has merits to the atmospheric chemistry community particularly who intends applying the CRM method for their field observations or using the dataset for the research. I would recommend the publication of this manuscript after addressing following points.

Reply: We would like to thank the reviewer for the insightful comments, which helped us tremendously in improving the quality of our work. Please find the response to individual comments below.

1. The reaction of CO with OH will produce $HO_2$, which ends up recycling OH in the excess NO environment. Please include the discussion in the manuscript.

Reply: Many thanks, we have included the discussion in the manuscript.

We added a sentence in the revised manuscript (Line 338-340): **Similarly, the produced $HO_2$ from the reactions of CO and $SO_2$ with OH will end up recycling OH in the excess NO environments and thus reduce the fitting slopes.**

2. It is certainly a good start to test the ICRM reactor with three VOCs but likely VOCs with various function groups required to evaluate. If the authors could suggest the list of compounds to be tested and provide justification, that would trigger follow up studies from other research groups.

Reply: We appreciate the reviewer for the comment. In addition to the four individual VOC species, we also calibrate using a mixed gas standard with 16 VOC species, namely acetaldehyde, methanol, ethanol, isoprene, acetone, acetonitrile, methyl vinyl ketone, methyl ethyl ketone, benzene, toluene, o-xylene, α-pinene, 1,2,4-trimethylbenzene, phenol, m-cresol, naphthalene. The calibration slope is close to those

of the three individual VOC species, indicating that the $RO_2$ + NO reactions for these investigated VOCs should be similar. In addition, we also calibrated methane in the revised manuscript and the results were added in Figure 4. Nevertheless, we agree with the reviewer that it is necessary to calibrate more VOC species in the future, especially considering that different VOCs species dominate in different environment, such as forest areas and various emission sources. For example, isoprene and terpenes have high reactivity contribution in forests and rural sites. Therefore, isoprene, α-Pinene and β-Pinene is suggested to be calibrated in the following study. Typical branched olefin, other aromatics (such as ethylbenzene) and important oxygenated VOCs (such as formaldehyde and glyoxal) should also be calibrated in the future.

We added the following sentences in the revised manuscript (Line 404-414): **In this study, we calibrated four individual representative VOC species (methane, propane, propene, toluene). In addition, we also calibrated the mixed standard gases with 16 VOC species including representative oxygenated VOCs (acetaldehyde, methanol, ethanol, acetone, acetonitrile, methyl vinyl ketone, methyl ethyl ketone), biogenic VOCs (isoprene, α-pinene), typical aromatics (benzene, toluene, o-xylene, 1,2,4-trimethylbenzene, naphthalene, phenol, m-cresol). The calibration slope is close to those of the four individual VOC species, indicating that the $RO_2$ + NO reactions for these investigated VOCs should be similar. Nevertheless, given that there are different VOCs compositions in different environment such as forest, urban area and emission sources, calibrations for more individual VOCs species might be also needed.**

[Figure]

52

53 Figure 4. The OH reactivity calibration of the improved CRM system using different

54 standard gases. (a) The calibrating results of organic species including methane,

55 propane, propene, toluene, and a mixture of 16 VOC species through arm C. (b) The

56 calibrating results of inorganic species including CO and $SO_2$. The measured OH

57 reactivity was calculated based on the C2 mode shown in Fig. 2 in the ICRM system.

---

## Author Comment (AC2) · 23 Jan 2021

**Responses to Reviewers**

**Reviewer 2**

The manuscript reported an improved CRM method to measure OH reactivity. The reactor of traditional CRM method was modified to suppress $HO_2$ formation. The lab experiment results were promising that the measured and calculated $k_{OH}$ by trace gases have good linearity. The ambient measurements showed that the new ICRM method could be used under urban ambient air. The manuscript is well within the scope of AMT. I recommend publication after attention to the following comments.

Reply: We are very grateful for all the detailed comments and the valuable suggestions, which helped us greatly in improving our manuscript. Please find the response to individual comments below.

General comments:

1. The reaction time of $NO+HO_2$ (or OH) in arm A should be clearly specified. HONO will be formed with no doubt and will it cause any interference with CRM?

Reply: (1) The initial $HO_2$ concentration is about 4 ppbv. The lifetime of $HO_2$ at 50 ppbv NO is at the time scale of 0.1 s, given that the reaction rate constant of $NO+HO_2$ is $8.0\times10^{-12}$ $cm^3$ $molecule^{-1}$ $s^{-1}$. The rection time of $NO+HO_2$ in arm A is estimated to be around ~ 0.1 s, during which most of $HO_2$ will be consumed. Hence there will be only a small fraction of $HO_2$ entering the main body of the reactor. (2) In the reactor, OH reacts with introduced or ambient NO to produce HONO, which can reproduce OH and NO by photolysis. As we have improved the structure of arm A to avoid UV light entering the main body of reactor, the photolysis of HONO is expected to be negligible. We pointed this out in line 478-480 of the revised manuscript: "**The change of the structure of arm A also ensures that the photolysis of $H_2O$, HONO, $NO_2$, and VOCs inside the ICRM reactor is weaker than that in the original CRM system.**"

We added the following sentences in the revised manuscript (Line 451-456): **Finally, the reaction time between $HO_2$ and NO should be noted. The initial $HO_2$ concentration is about 4 ppbv. The lifetime of $HO_2$ at 50 ppbv NO is at the time**

**scale of 0.1 s, given that the reaction rate constant of NO+HO$_2$ is 8.1×10$^{-12}$ cm$^3$ molecule$^{-1}$ s$^{-1}$ (Sander, 2006). The rection time of NO+HO$_2$ in arm A is estimated to be around 0.1 s, during which most of HO$_2$ will be consumed. Hence there will be only a small fraction of HO$_2$ entering the main body of the reactor.**

We added the following sentences in the revised manuscript (Line 481-484): **In this system, OH reacts with introduced NO or ambient NO to produce HONO, which can reproduce OH and NO by photolysis. As we have improved the structure of arm A to avoid UV light entering main body of the reactor, photolysis of HONO is expected to be low.**

2. In section 3.1, the NO addition measurement showed that Pyrrole concentration decreased to the minimum with NO around 40 ~ 50 ppbv (Fig.2). I think the increase of Pyrrole with NO is due to NO+OH reaction which lower the produced OH concentration.

Reply: Yes, we agree with you that the increase of pyrrole with NO is due to NO+OH reaction which lower the produced OH concentration.

Line 273-274: **As the NO concentration exceeds 50 ppbv, pyrrole concentrations increase again, due to large excess NO competes with pyrrole for reaction with OH radicals.**

3. In section 3.2 and Fig.4, the authors gave very promising dataset of the measured and calculated k$_{OH}$. The linearity were all very good but the slope were not close to 1. Since the ICRM method introduced extra NO in the reactor, the cycling of OH-HO$_2$-OH can not be avoid due to the reaction time in the reactor. The influence of initial HO$_2$ was suppressed and at mean time the HOx cycling was enhanced. I would guess the slopes of CO and VOCs in Fig. 4 are related to this issue. Probably more VOCs should be tested before application in ambient air.

Reply: Yes, as the ICRM method introduced extra NO in the reactor, the cycling of OH-HO$_2$-OH cannot be avoid due to the reaction time (~11s) in the reactor, and the recycled OH from $RO_2$ will deplete pyrrole thus leading to a $R_{meas}$ lower than the $R_{true}$.

We agree with the reviewer that the slopes of CO and VOCs lower than one in Fig. 4

are related to this issue. We also calibrate methane in the revised manuscript and similar results are shown (Figure 4a). In addition to the four individual VOC species, we also calibrated the mixed standard gases with 16 VOC species. The calibration slope is close to those of the three individual VOC species, indicating that the $RO_2$ + NO reactions for these investigated VOCs are similar. Nevertheless, we agree with the reviewer that it is necessary to calibrate more VOC species in the future, especially considering that different VOCs species dominate in different environment, such as forest areas and various emission sources. For example, isoprene and terpenes have high reactivity contribution in forests and rural sites. Therefore, isoprene, α-Pinene and β-Pinene is suggested to be calibrated in the following study. Typical branched olefin, other aromatics (such as ethylbenzene) and important oxygenated VOCs (such as formaldehyde and glyoxal) should also be calibrated in the future.

We added the following sentences in the revised manuscript (Line 404-414): **In**

**this study, we calibrated four individual representative VOC species (methane,**

**propane, propene, toluene). In addition, we also calibrated the mixed standard**

**gases with 16 VOC species including representative oxygenated VOCs**

**(acetaldehyde, methanol, ethanol, acetone, acetonitrile, methyl vinyl ketone,**

**methyl ethyl ketone), biogenic VOCs (isoprene, α-pinene), typical aromatics**

**(benzene, toluene, o-xylene, 1,2,4-trimethylbenzene, naphthalene, phenol, m-**

**cresol). The calibration slope is close to those of the four individual VOC species,**

**indicating that the $RO_2$ + NO reactions for these investigated VOCs should be**

**similar. Nevertheless, given that there are different VOCs compositions in**

**different environment such as forest, urban area and emission sources,**

**calibrations for more individual VOCs species might be also needed.**

[Figure]

Figure 4. The OH reactivity calibration of the improved CRM system using different standard gases. (a) The calibrating results of organic species including methane, propane, propene, toluene, and mixture gases of 16 VOC species through arm C. (b) The calibrating results of inorganic species including CO and SO2. The measured OH reactivity was calculated based on the C2 mode shown in Fig. 2 in the ICRM system.

Specific comments:

Line 58: Better to include NO in the equation.

Reply:We thank the reviewer for the careful comment. We have modified equation 1 in the revised manuscript (Line 58):

$$R_{OH} = k_{CO}[CO] + k_{NO}[NO] + k_{NO_2}[NO_2] + k_{SO_2}[SO_2] + k_{O_3}[O_3] + \sum_i^n k_{VOC_i}[VOC_i]$$

$$(1)$$

Line 131: Please specify the brand and type of the lamp, as well as its emission line.

Reply: The brand and type of UV lamp that we used is Analytik Jean (type: 90-

0012-01), and its emission line is 254 nm.

We added the following sentences in the revised manuscript (Line 132): **Arm A**

**consists of a pen-ray spectral mercury lamp (Analytik Jean; 90-0012-01), over**

**which nitrogen (humidified or dry) is passed through arm B at a constant flow**

**rate.**

Line 158: "An underlying assumption of the CRM approach is that the influence of the species in ambient air on OH radicals in the reactor is ignorable." The sentence is ambiguous. It is also useful to give the theoretical OH mixing ratio in the reactor here.

Reply: We thank the reviewer for the comment. The species in ambient air will of course influence the concentration of OH radicals by reacting with OH radicals. We mean that an underlying assumption of the CRM approach is that the production of OH

radicals is just from the photolysis of $H_2O$ under UV lamp, and the influence of the species in ambient air on the production of OH radicals in the reactor is ignorable. The theoretical OH mixing ratio in the original reactor is about $5 \sim 20$ ppbv, which depends on the introduced pyrrole concentration to ensure the pyrrole/OH ratio is 2:1~3:1. For

ICRM, the total OH radical concentration including production from UV lamp and from the reaction of $HO_2$ with NO is about 10 ppbv.

We added the following sentences in the revised manuscript (Line 160-164): **An**

**underlying assumption of the CRM approach is that the influence of the species in**

**ambient air on the production of OH radicals in the reactor is ignorable. The**

**theoretical OH mixing ratio in the original CRM reactor is about $5 \sim 20$ ppbv,**

**which depends on the introduced pyrrole concentration to ensure the Pyrrole/OH**

**ratio is 2:1~3:1.**

We added also the following sentences in the revised manuscript (Line 284-287):

**Under this optimized condition, the pyrrole concentration decreased to 12.3 ppbv,**

**indicating the total OH radical concentration including production from UV lamp**

**and from the reaction of $HO_2$ with NO is about 10 ppbv in the ICRM system.**

Line 181: Did the author try different structure (length, ID. etc) of arm A to get an optimal setup?

Reply: (1) The length of arm A will determine initial OH concentration passing into the reactor and the reaction time of $HO_2$ with NO. The longer arm A is beneficial for longer reaction time of $HO_2$ with NO, but lower OH concentrations passing into the reactor due to wall loss. We chose an appropriate length of arm A (12 cm) to ensure appropriate OH concentration (4 ppbv) and reaction time of $HO_2$ with NO (~0.1 s). (2) For ID, arm A consists of two sections of glass tube with 1/2 inch OD (ID: 0.62 cm, length: 7 cm) and one 1/4 inch OD (ID: 0.32 cm, length: 5 cm) respectively. This ensure that UV light is mostly confined in 1/2 inch OD glass tube of arm A, as the glass tube is constructed with decreasing diameter following the direction of gas flow. However, the reaction time of $HO_2$ with NO in arm A is very short (~0.1 s), which needs to be solved in the future.

We modified the following sentences in the revised manuscript (Line 182-193): **Arm A consists of one 1/2 inch OD (ID: 0.62 cm, length: 7 cm) glass tube and one 1/4 inch OD (ID: 0.32 cm, length: 5 cm) glass tube. The longer arm A is beneficial for longer reaction time of $HO_2$ with NO, but lower OH concentrations passing into the reactor due to wall loss. We chose an appropriate length of arm A (12 cm) to ensure appropriate OH concentration (4 ppbv) and reaction time of $HO_2$ with NO (~ 0.1 s). The purpose of the two-section structure is to ensure that the UV light is mostly confined within a 1/2 inch OD glass tube of arm A, as the diameter of arm A goes from wide to slender. The new structure of arm A leads to lower OH concentrations (decreased by approximately 50%) passing into reactor compared with the original CRM system due to wall loss, but OH radicals produced from the reaction of $HO_2$ radicals with NO can partially compensate for this loss.**

We added the following sentences in the revised manuscript (Line 468-475): **Arm A consists of two section of glass tube with 1/2 inch OD and 1/4 inch OD, respectively (Fig. 1c). UV light is mostly confined in 1/2 inch OD glass tube of arm A, as the glass tube is constructed with decreasing diameter following the direction**

**of gas flow. This reduces the amount of UV light getting into the main reaction part of the reactor. The improved structure of arm A leads to lower OH concentrations (decreased by approximately 50%) passing into reactor due to wall loss, but the OH radicals produced from the reaction of HO$_2$ radicals with NO can partially compensate for this loss.**

Line 250: It is better to include OH+NO reaction here. Is this reaction also include in the box model?

Reply: Thank you for pointing this out. We have included it. Yes, this reaction was included in the box model for the simulation.

Line 263: $$NO + OH \rightarrow HONO \qquad\qquad R8$$

Line 283: the rate constant of OH+NO should be given here or in Fig.3, when calculated R-true of NO.

Reply: Corrected. We have added the reaction rate constant of OH+NO (Line 297).

We modified the following sentences in the revised manuscript (Line 297-298):

**Measured OH reactivity of NO ($k_{NO}$ = 9.7×10$^{-12}$ cm$^3$ molecule$^{-1}$ s$^{-1}$ according to IUPAC lasted evaluation in November 2017) agreed well with the corresponding true values.**

Line 404: The rate constant were quoted from Atkinson 2004, which is a well-known reference. I would suggest the authors also check the new evaluations or recommendations on JPL-2015 or IPUAC sources.

Reply: Thank you for providing us with the useful reference. We have updated the calculated reactivity data by using reaction coefficients from IUPAC sources (http://iupac.pole-ether.fr). For reactions that are unavailable in IUPAC, we used reaction coefficients from JPL-2015 evaluation.

**Other modifications:**

(1) Additionally, we added the detection limit in the revised manuscript.

Line 535-538: **The detection limit of ICRM was determined to be 2.3 s$^{-1}$ for an averaged pyrrole-to-OH ratio of 2.3 according to the method proposed by Michoud et al. (2015) (Fig S9). This means that the minimum detection limit for the reactivity of sample air would be about 5 s$^{-1}$ (typically diluted in the glass reactor by a factor 2).**

(2) We revised the fitting results between $R_{true}$ and $R'_{meas}$ ($R_{true} - R'_{meas}$) increases with NO concentrations for different VOC species and different reactivity levels (Fig. 5 and Line 354-356 in the revised manuscript).

Line 355-357: **Similar to previous study (Michoud et al., 2015; Praplan et al., 2017; Yang et al., 2017), the difference between $R_{true}$ and $R'_{meas}$ ($R_{true} - R'_{meas}$) increases with NO concentrations for different VOC species and different reactivity levels.**

(3) In Figure S2, the solid line is replaced by dots, which is more reasonable for the expression of this Figure.

[Figure]

**Figure S2**. The remaining concentrations of pyrrole, NO, $NO_2$, $HO_2$, and $RO_2$ outflowing of the reactor (with the reaction time of ~ 11 s) as a function of introduced NO in the reactor.

[Figure]

Figure S9. Total OH reactivity detection limit measured for the ICRM.

**References**

Michoud, V., Hansen, R. F., Locoge, N., Stevens, P. S., and Dusanter, S.: Detailed characterizations of the new Mines Douai comparative reactivity method instrument via laboratory experiments and modeling, Atmospheric Measurement Techniques, 8, 3537-3553, doi: 10.5194/amt-8-3537-2015, 2015.

Praplan, A. P., Pfannerstill, E. Y., Williams, J., and Hellén, H.: OH reactivity of the urban air in Helsinki, Finland, during winter, Atmospheric Environment, 169, 150-161, doi: 10.1016/j.atmosenv.2017.09.013, 2017.

Sander, S. P., B. J. Finlayson-Pitts, R. R. Friedl, D. M. Golden, R. E. Huie, H. Keller-Rudek, C. E. Kolb, M. J. Kurylo, M. J. Molina, G. K. Moortgat, V. L. Orkin, A. R. Ravishankara and P. H. Wine: Chemical Kinetics and Photochemical Data for Use in Atmospheric Studies Evaluation Number 15, JPL Publication 06-2, Jet Propulsion Laboratory, 2006.

Yang, Y., Shao, M., Keßel, S., Li, Y., Lu, K., Lu, S., Williams, J., Zhang, Y., Zeng, L., Nölscher, A. C., Wu, Y., Wang, X., and Zheng, J.: How the OH reactivity affects the ozone production efficiency: case studies in Beijing and Heshan, China, Atmospheric Chemistry and Physics, 17, 7127-7142, doi: 10.5194/acp-17-7127-2017, 2017.